# Risk of Postoperative Bleeding in Tonsillectomy for Peritonsillar Abscess, as Opposed to in Recurrent and Chronic Tonsillitis—A Retrospective Study

**DOI:** 10.3390/ijerph18041946

**Published:** 2021-02-17

**Authors:** David Slouka, Štěpánka Čejková, Jana Hanáková, Petr Hrabačka, Stanislav Kormunda, David Kalfeřt, Alena Skálová, Václav Šimánek, Radek Kucera

**Affiliations:** 1Department of Otorhinolaryngology, University Hospital in Pilsen, Faculty of Medicine in Pilsen, Charles University, 305 99 Pilsen, Czech Republic; slouka@fnplzen.cz (D.S.); cejkovas@fnplzen.cz (Š.Č.); hanakovaj@fnplzen.cz (J.H.); hrabackap@fnplzen.cz (P.H.); skormunda@centrum.cz (S.K.); 2Department of Otorhinolaryngology, Head and Neck Surgery, 1st Faculty of Medicine, Charles University and University Hospital Motol, 150 06 Praha 6, Czech Republic; david.kalfert@fnmotol.cz; 3Department of Pathology, University Hospital in Pilsen, Faculty of Medicine in Pilsen, Charles University, 305 99 Pilsen, Czech Republic; skalova@fnplzen.cz; 4Department of Immunochemistry Diagnostics, University Hospital in Pilsen, Faculty of Medicine in Pilsen, Charles University, E. Benese 1128/13, 305 99 Pilsen, Czech Republic; KUCERAR@fnplzen.cz; 5Faculty of Medicine in Pilsen, Institute of Pharmacology and Toxicology, Charles University, 323 00 Pilsen, Czech Republic

**Keywords:** peritonsillar abscess, chronic tonsillitis, recurrent tonsillitis, postoperative bleeding, personalized treatment

## Abstract

Tonsillectomy is a routine surgery in otorhinolaryngology and the occurrence of postoperative bleeding is not a rare complication. The aim of this retrospective, observational, analytic, cohort study is to compare the incidence of this complication for the most common indications. A group of patients indicated for tonsillectomies for peritonsillar abscess (group I) was compared to a group of patients indicated for chronic and recurrent tonsillitis (group II). There are a lot of pathophysiological differences in patients indicated for acute tonsillectomy for peritonsillar abscess and in patients indicated for elective tonsillectomy for chronic or recurrent tonsillitis. No technique to minimize the risk of bleeding after tonsillectomy has been found and a large part of postoperative bleeding occurs in postoperative home-care, which makes this issue topical. In total, 2842 unilateral tonsillectomies from the years 2014–2019 were included in the study. Bleeding occurred in 10.03% and, surprisingly, despite completely different conditions in the field of surgery (oedema, acute inflammation in peritonsillar abscess), there was no statistically significant difference between incidence of postoperative bleeding in the studied groups (*p* = 0.9920). The highest incidence of bleeding was found in the patients of group I on the eighth postoperative day, with those aged 20–24 years (*p* = 0.0235) being the most at risk, and in group II, on the sixth postoperative day, with those aged 25–29 years (*p* = 0.0128) and 45–49 years (*p* = 0.0249) being the most at risk.

## 1. Introduction

Due to a wide range of indications [1,2], tonsillectomy (TE) is one of the most common surgical procedures in ENT departments. Tonsillectomy is classified as a minor operation, yet various complications associated with these operations are quite common [3,4]. We divide complications into perioperative and postoperative categories. Perioperative complications include: bleeding, anesthesiology complications, injury to surrounding structures, the retention of tonsil tissue residues, etc. Postoperative complications include: bleeding, dysphagic problems (sometimes associated with insufficient food intake), tonsillar fossa infection, velopharyngeal dysfunction, chronic hypertrophic pharyngitis, voice changes, taste disturbances, but also relatively rare complications such as torticollis, deep throat inflammation, pneumonia, emphysema, etc. [5,6].

Postoperative bleeding after tonsillectomy is one of the most common and feared complications [7]. The available literature indicates its occurrence as 2–15% [8,9], and the interval when post-tonsillectomy hemorrhage (PTH) can occur is very wide: between the 1–21 postoperative days [10,11]. Early bleeding in the initial postoperative days occurs during hospitalization, whereas late bleeding occurs when the patient is already in their home environment and therefore represents a higher risk [3,12]. 

The most common indications for tonsillectomy include chronic tonsillitis (CHT), recurrent tonsillitis (RT) and peritonsillar abscess (PTA) [2,3]. CHT and RT can be diagnosed in patients of almost all ages, but they are rare in old age people and in the vast majority of cases, the process is two-sided [13,14]. Peritonsillar abscess is typically unilateral— it is relatively rare on both sides [15,16]. PTA is the most common local complication of acute tonsillitis. It affects all age groups, including senior patients, albeit less often than younger patients [17]. 

The aim of the study was to determine whether the indication for acute tonsillectomy in PTA (different pathophysiological conditions—oedema, acute inflammation in tissue, more difficult orientation in the surgery field) is associated with a higher risk of postoperative bleeding than elective tonsillectomy indicated for the treatment of CHT or RT. The resulting data were evaluated not only epidemiologically, but also with the intention of personalizing the treatment process, as this could help prevent the emergence of life-threatening postoperative complications. 

## 2. Material and Methods

### 2.1. Group of Patients

Data of the total 2340 patients who underwent tonsillectomy in 2014–2019 in our department were explored. Patients were divided according to the diagnosis that was an indication for tonsillectomy. Patients who did not meet the inclusion criteria were excluded (flow chart, Figure 1). All the patients underwent surgery with no exacerbation of comorbidities (hypertension, diabetes mellitus, etc.) and postoperative pain was managed using analgesics that do not raise the risk of PTH: paracetamol and metamizol.

A total of 1994 patients were enrolled in our retrospective, observational, analytic, cohort monocentric study. The group includes 980 (49.15%) women and 1014 (50.85%) men for whom tonsillectomy was indicated in 2014–2019. Patients’ ages ranged from 2 to 86 years. Patients were divided into 2 groups according to the indicative diagnosis for TE (group I for peritonsillar abscess, group II for chronic or recurrent tonsilitis). The distribution of patients by age and diagnostic group is shown in Figure 2.

Details of the demographic distribution of the group are given in Table 1. There were 1100 patients in group I and 894 patients in group II: 650 patients (276 men (42.46%), 374 women (57.54%)) were indicated for CHT; 244 patients (96 men (39.34%), 148 women (60.66%)) were indicated for RT. 

### 2.2. Methods

For the purposes of the study, the data of a total of 2842 unilateral tonsillectomies (resected tonsils) were included in the study group, of which 1146 tonsillectomies were performed primarily as a unilateral procedure and 1742 tonsillectomies were performed as part of bilateral procedures. All the tonsillectomies were performed by board-certified doctors or under the supervision of a board-certified doctor. 

There were 1100 patients in group I and all of these patients underwent unilateral tonsillectomy on the abscess-side; group I contains 1100 unilateral tonsillectomies (1100 resected tonsils, which is 38.71% of the whole study). 

In group II there were 894 patients in total and 1742 resected tonsils (61.29% of the whole study). A total of 650 patients (276 men (42.46%) and 374 women (57.54%)) were indicated for CHT in group II. The sample included 1262 resected tonsils (44.40% of the whole study). A total of 244 patients (96 men (39.34%) and 148 women (60.66%)) were indicated for RT in group II, comprising 480 resected tonsils (16.89% of the whole cohort). 

Type of surgery distribution is presented in Table 2.

The inclusion criteria were: extracapsular tonsillectomy under general anesthesia and indication for TE for PTA, CHT, RT, and cold-steel technique. Exclusion criteria: tonsillectomy under local anesthesia; use of a surgical technique other than conventional cold-steel; TE indicated in other major procedures (snoring, sleep apnea syndrome); parapharyngeal abscess or deep neck abscess; patients with blood clotting disorders; patients in preparation for transplantation; poor cooperation—the signing of a waiver by the patient in the postoperative period; oncological indications including diagnostic TE in case of tumor suspicion; surgery of the processus styloideus elongatus; use of antiplatelet agents; use of anticoagulants; bilateral tonsillectomy for PTA.

Patients with uncomplicated courses of healing were monitored by an outpatient ENT and their GP once a week during the home-care period, with the GP deciding when they could go back to work. The total duration of sick leave and follow-up lasted two to three weeks. The patients in the PTH group were individually monitored in the home-care period in the outpatient consulting room of the ENT department where the surgery was performed. The maximum duration of a follow-up was 5 weeks. The hospitalization for PTH lasted from 3 days in observed patients to 2 weeks in complicated cases. 

Ethical Approval: Informed consent was obtained from all the participants. The study was approved by the Ethical Committee of the University Hospital in Pilsen on 8 July 2013. 

### 2.3. Methodology of Stopping Bleeding

In the study, all patients with postoperative bleeding suffered from PTH on one side and were hospitalized, even if the data were only anamnestic. Anamnestic and minor insignificant bleeding were addressed by observation and application of hemostyptics. Tonsillar fossa compression using a gauze tampon and bipolar coagulation under local anesthesia were used to control significant bleeding. Electrocoagulation under general anesthesia was indicated in cases of unsuccessful treatment of bleeding with local anesthesia, or in cases of massive bleeding. Other methods (suture of palatal arches, hypopharyngeal tamponade) were not used.

### 2.4. Statistical Methods

Statistical analysis was performed using SW SAS (Cary, NC, USA) and SW STATISTICA (StatSoft, Inc., Tulsa, OK, USA). The descriptive statistics, such as frequencies, mean, standard deviation, variance, median, minimum and maximum were calculated to describe the investigated parameters. Box plots, pie charts and histograms were used for graphical presentation of the data. The nonparametric ANOVA tests (Wilcoxon rank-sum test and median test) were used for comparisons of age and number of bleedings between studied groups. Chi-square test or Fisher’s exact test were used to compare frequencies. The clinical impact of factors such as sex, side of surgery, age and seasons were evaluated using odds ratio calculations. Time of bleeding after surgery was calculated using Kaplan-Meier survival and the Log-rank test was used for comparison. Risk function was used to calculate the risk of bleeding on particular days after surgery. Statistical significance was determined at the level of 5%.

## 3. Results

Among the examined groups of diagnoses I (PTA) and II (CHT + RT), there was no statistically significant difference in the frequency of postoperative bleeding (*p* = 0.9920). The PTH occurred on one side in all cases.

The general risk factor in the whole group was male gender (*p* = 0.0104). No difference was found between the genders in the incidence of the observed complication within diagnosis group I (PTA)—neither for the left nor for the right side. In the examined group of diagnoses II (CHT + RT), there was no difference in the incidence of bleeding on the right and left sides of the operation, but a statistically significant difference was found for the gender subgroups: men had a higher risk of bleeding (*p* = 0.0129). For details, see Table 3 and Table 4.

An analysis of the relative frequency of bleeding did not show an increased incidence of complications in any of the months during the year: group I (PTA) *p* = 0.8827, group II (CH + RT) *p* = 0.6737. An evaluation of the finding at the time of examination, degree of bleeding and management of bleeding did not yield a statistically significant difference (*p* = 0.3045). The examination of repeated PTH was without significant differences in either group (*p* = 0.5774). For details, see Table 5 and Table 6.

There was no statistically significant difference in age between the groups of patients with and without postoperative bleeding within both groups (*p* = 0.1656). Results are presented in graphic form in Figure 3a,b.

When dividing the examined groups (I and II) into subgroups after five years, three groups appeared that were at a statistically significant higher risk of bleeding. For group I (PTA), the age group most at risk was 20–24 year-old patients (*p* = 0.0235) and for group II (CHT + RT), those aged 25–29 years (*p* = 0.0128) and 45-49 years (*p* = 0.0249) were at the most risk. For details see Table 7.

The probability of bleeding on individual days is shown in Figure 4. Here, we proved that the eighth postoperative day was the riskiest for the PTA group, followed by the sixth postoperative day for the CHT + RT group. In the sample of PTH, bleeding occurred in 173 cases (60.7%) in the home-care period of post-tonsillectomy care (for PTA *n* = 64, 59.81%; for CHT + RT *n* = 109, 61.23%). 

## 4. Discussion

### 4.1. Differences in the Course of PTA and CHT + RT

Due to their placement, the palatal tonsils are closely related to large blood vessels and the respiratory tract, so some complications can endanger the patient’s life. All patients with postoperative bleeding after tonsillectomy must be hospitalized, even if there is only a history of bleeding. Conservative treatment steps such as observation, tonsillar fossa compression using a gauze tampon, application of hemostyptics and invasive steps such as electrocoagulation of source vessels under local or general anesthesia, revision of the tonsillar bed after tonsillectomy and suture of palatal arches are used as solutions for this complication [18,19]. In the rare cases of dramatic, life-threatening bleeding, blood replacement is part of the therapy [20]. Such life-threatening bleeding was not observed in our study. Our study investigated the idea of different risk levels of postoperative bleeding after tonsillectomy for PTA, CHT or RT because the maximum inflammatory and repair processes in CHT, as well as in RT, occur in tonsil tissue and bilaterally, with only a minimum of these processes taking place in the surrounding tissue of tonsillar fossa; the basic anatomical conditions remain intact. On the contrary, in PTA, the process is most often one-sided and its impact on the surrounding tissues can be detrimental. First, a pe-ritonsillar phlegmon develops, then a pyogenic membrane forms and the inflammatory process is bound by an abscess [21,22]. Swelling, as well as the presence of the abscess and the surrounding phlegmon, makes orientation in the surgical field significantly more complicated than in tonsillectomy indicated for CHT or RT. The incidence of left and right PTA involvement is reported to be approximately the same in the literature [23,24]. Unlike CHT and RT, due to the pathophysiology of the PTA process, it is often complicated by the development of other serious complications such as airway obstruction [25] or deep throat infections [26,27]. The incidence of PTA occurrence in the population is 1045/100,000 inhabitants/year [21,22]; there are studies that report the presence of chronic tonsillitis in as much as 10–25% of the population [28,29], and RT in 12% of the population [30,31]. 

### 4.2. Postoperative Bleeding

Studies on the topic of bleeding after tonsillectomy are most often focused on surgical technique. They often compare the results of intracapsular and extracapsular TEs. These works report a lower risk of postoperative bleeding after intracapsular surgery. This technique, however, is suitable for the treatment of tonsil hypertrophy, but not for the treatment of chronic inflammation or even PTA [32,33]. The topic of extracapsular operations involves the further discussion of “alternative techniques” such as: various types of lasers [34,35], harmonic scalpel [36], coblation [37] and radiofrequency [38]. Some published studies report a reduction in the incidence of perioperative bleeding [39,40], but essentially the same frequency of postoperative bleeding [41,42,43]. In accordance with the vast majority of other ENT departments, our department prefers extracapsular type TE and the cold-steel design [44,45]. All tonsillectomies included in the study correspond to this. The work of Frycova et al. from 2012 lists the left lymph tissue residue after TE as a possible factor and source of postoperative bleeding [46]. This cause was not found in our file. The patients whose cases were standard were dismissed on the fifth postoperative day in our study. The range of postoperative bleeding after tonsillectomy is relatively wide in the literature and ranges between 1 and 21 postoperative days [47,48,49], which corresponds to our results. In our published work, the overall average incidence of postoperative bleeding was 10.03% (9.73% for group I (PTA) and 10.22% for group II (CHT + RT)), which also corresponds to the literature data.

### 4.3. Age, Gender, Season and Side Dependence

The aim of our study was to compare the risks of postoperative bleeding in a group of patients who underwent acute tonsillectomy for PTA (with many pathophysiological differences such as oedema, acute inflammation in tissue, more difficult orientation in the surgery field, etc.) with the risks of bleeding in a group of patients indicated for elective TE due to CHT or RT, as well as in subgroups of patients defined by age, sex and season. In practice, the vast majority of tonsillectomies for CHT and RT are bilateral and, conversely, the vast majority of tonsillectomies for PTA are unilateral. For the purposes of preserving the homogeneity of the compared samples and in order to compare the procedures (PTA versus CHT + RT), tonsillectomy was defined as a unilateral operation (one resected tonsil) and tonsillectomies derived from bilateral procedures were statistically evaluated as two separate tonsillectomies (right-sided and left-sided). The incidence of bleeding in the postoperative period was almost the same in both groups studied by us: group I (PTA)—9.73%, and group II (CHT + RT)—10.22% (*p* = 0.9920). Postoperative bleeding in our study did not show a time course dependence in either group: for I (PTA)—*p* = 0.827; for II (CH + RT)—*p* = 0.6737. In the study of the whole group, a dependence on gender was recorded: the group most at risk was men (*p* = 0.0104). The study of Ikoma et al. from 2014 employed a similar methodology and produced PTH results only slightly higher than ours (incidence of PTH—11.6%) [50] when dealing with PTH after TE for CHT. Ikoma says that risk factors for PTH were adult age and male gender. The fact that males have a stronger tendency to bleed postoperatively has been published in the literature in the past [51]. However, no clear explanation has been given for this fact.

In our work, after dividing the studied groups into five-year periods, we demonstrated that the PTA group included a subgroup that was more at risk: 20–24 years (*p* = 0.0235); men have twice the risk of bleeding than women (*p* = 0.0329); two age groups were found to be more at risk in the CHT + RT group: 25–29 years (*p* = 0.0128) and 45–49 years (*p* = 0.0249), with the 25–29 year-old men are being twice as likely to bleed. The work of Kim et al. from 2012 [8] on a similarly large group of tonsillectomies for CHT + RT (*n* = 2254) had an overall incidence of post-TE bleeding that was lower than other published studies (only *3,5*%) and reported a higher incidence of bleeding in people over 65 years (*p* < 0.0001).

The side that was operated on did not play a role in the groups compared in our study. Giger et al. 2005, with a partially different methodology (bilateral TEs were performed for PTA and they had a significantly smaller group, *n* = 205), indicates an overall incidence of postoperative bleeding that is higher than in our group (13%) and 2x higher for the contralateral side versus the abscess side. They list male sex (*p* = 0.042) and bilateral TE in PTA (*p* = 0.020) as risk factors [50]. The conclusions of the study are consistent with our hypothesis of a higher risk of bleeding for patients with PTA than with elective TE. Giger subsequently recommends performing only ipsilateral TE for PTA. We have been practicing this for a long time, but in our work, in contrast to Giger’s results, we have demonstrated the occurrence of PTH as less frequent. Despite significantly different conditions of acute TE for PTA (swelling, acute inflammation) versus planned surgery for CHT or RT, the differences in risk of PTH between the two groups were surprisingly small in our study. The difference in frequency of postoperative bleeding was not statistically significant (*p* = 0.9920). There was also a statistically insignificant difference in the individual types of observed bleeding (*p* = 0.7524), the approach to their solution (*p* = 0.3045), and even the incidence of recurrent postoperative bleeding (*p* = 0.5774). In accordance with the interval given in the literature [49,50,51], the graph of the probability of bleeding (Figure 4) shows that in our study the riskiest postoperative day was the eighth (probability of bleeding 1.74%) for the PTA group, for the CHT + RT group the riskiest day was the sixth postoperative day (1.62% probability of bleeding). The bleeding on those days was experienced in home-care, which increases the danger associated with bleeding.

A limitation of our study may be its monocentricity and the fact that the postoperative diet was not monitored. The first limit is balanced by the advantage of having a large group of patients operated on (2842 resected tonsils for a period of 5 years). The postoperative diet in the inpatient period was the same for all patients who had undergone tonsillectomy. All the patients were educated regarding their diet during the following home-care period and were informed of all possible complications.

## 5. Conclusions

Tonsillectomy is undoubtedly a routine operation in otorhinolaryngology; the occurrence of postoperative bleeding is not, however, a rare complication. Techniques to minimize the risk of bleeding have not yet been found. Despite the widely different conditions in the field of tonsillectomy surgery for PTA, our work demonstrated an almost identical risk of postoperative bleeding in the indication for tonsillectomy as a treatment of PTA and in the indication for CHT + RT. Among the biggest risks resulting from tonsillectomy is the fact that the riskiest days of PTH fall into the home-care period of post-tonsillectomy treatment. In our sample, PTH occurred in more than 60% of patients in the home environment.

## Figures and Tables

**Figure 1 ijerph-18-01946-f001:**
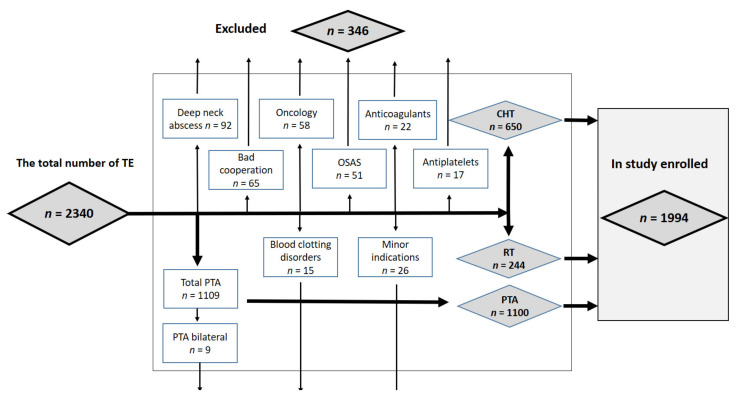
Flow chart—demonstration of an enroll/exclusion process.

**Figure 2 ijerph-18-01946-f002:**
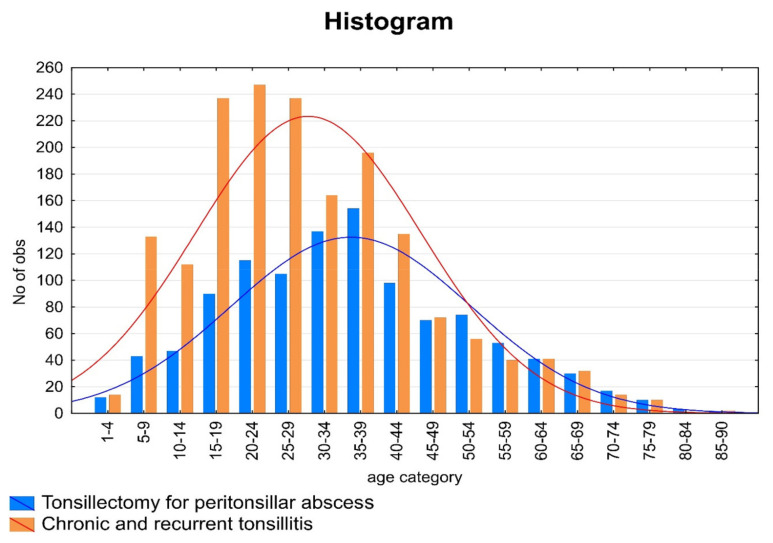
The age distribution of patients in both groups.

**Figure 3 ijerph-18-01946-f003:**
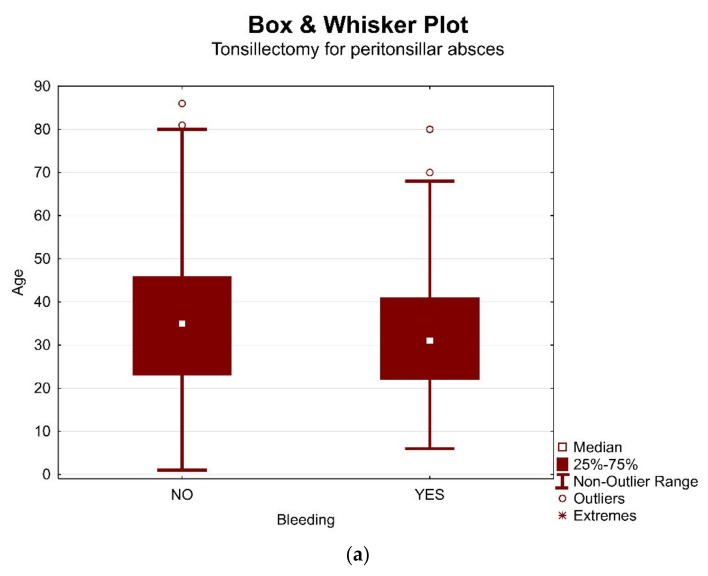
(**a**). Distribution of patients and bleeding depending on age in group I (peritonsillar abscess (PTA)). (**b**). Distribution of patients and bleeding depending on age in group II (chronic tonsillitis (CHT) + recurrent tonsillitis (RT)).

**Figure 4 ijerph-18-01946-f004:**
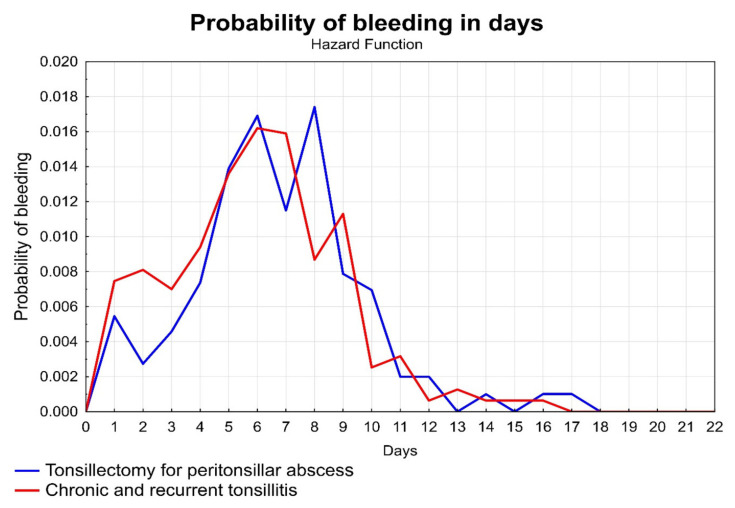
Probability of bleeding on individual days.

**Table 1 ijerph-18-01946-t001:** Age and gender characteristics of the patient group.

Diagnosis	Total		Gender					Age		
	Patients	%	Men	%	Women	%	*p*-Value M/W	Mean	Median	*p*-Value
I (PTA)	1100	55.17	642	58.36	458	41.64	**<0.0001**	35.41	35	**<0.0001**
II (CHT + RT)	894	44.83	372	41.61	522	58.39		29.54	27	
total	1994	100	1014	50.85	980	49.15	-	31.81	30	-

**Table 2 ijerph-18-01946-t002:** Type of surgery in the patient group.

Diagnosis	Tonsillectomy						Side of Tonsillectomy				
	Total	%	Bilateral	%	Unilateral	%	Left	%	Right	%	*p*-Value
I (PTA)	1100	38.71	0	0	1100	100	554	50.36	546	49.64	0.9920
II (CHT + RT)	1742	61.29	848	97.36	46	2.64	877	50.34	865	49.66	
total	2842	100	848	59.68	1146	40.32	1431	50.35	1411	49.65	-

**Table 3 ijerph-18-01946-t003:** Postoperative bleeding: men versus women.

	*n*	PTH	%	*p*-Value	Men			Women			*p*-Value
					Total	PTH	%	Total	PTH	%	
I (PTA)	1100	107	9.73	0.9920	642	68	10.59	458	39	8.52	0.2519
II (CHT + RT)	1742	178	10.22		729	90	12.35	1013	88	8.69	**0.0129**
total	2842	285	10.03	-	1371	158	11.52	1471	127	8.63	**0.0104**

**Table 4 ijerph-18-01946-t004:** Postoperative bleeding: right versus left sides.

	*n*	PTH	Right	%	Left	%	*p*-Value
I (PTA)	1100	107	56	10.26	51	9.21	0.5566
II (CHT + RT)	1742	178	93	10.75	85	9.69	0.4655
total	2842	285	149	52.28	136	47.72	0.3487

**Table 5 ijerph-18-01946-t005:** Finding at the time of examination and degree and management of bleeding.

Diagnosis	Finding of Bleeding					Degree of Bleeding and Therapy						
	Anamnestic or Insignificant		Significant			Observation and Hemostyptics		Compression and/or Coagulation - Local Anesthesia		Compression and/or Coagulation - General Anesthesia		
	*n*	%	*n*	%	*p*-Value	*n*	%	*n*	%	*n*	%	*p*-Value
I (PTA)	20	18.69	87	81.31	0.7524	76	71.03	7	6.54	24	22.43	0.3045
II (CHT + RT)	36	20.22	142	79.78		129	72.47	5	2.81	44	24.72	
total	56	19.65	229	80.35	-	205	71.93	12	4.21	68	23.86	-

**Table 6 ijerph-18-01946-t006:** Repeated postoperative bleeding in the two groups.

	1. PTH		2. PTH		3. PTH		
	*n*	%	*n*	%	*n*	%	*p*-Value
I (PTA)	107	9.73	11	11.28	1	0.93	0.5774
II (CHT + RT)	178	10.22	21	11.80	2	1.12	
total	285	10.03	32	11.23	3	1.05	-

**Table 7 ijerph-18-01946-t007:** Risk of bleeding in subgroups by age of patients

	Group I (PTA)				Group II (CHT + RT)			
Age	Total	PTH	%	*p*-Value	Total	PTH	%	*p*-Value
0–4	12	0	0	0.6195	14	1	7.14	1.0000
5–9	43	3	6.98	0.7919	133	8	6.02	0.0959
10–14	47	4	8.51	1.0000	112	8	7.14	0.2666
15–19	90	8	8.89	0.7794	237	19	8.02	0.2287
20–24	115	18	15.65	**0.0235**	247	27	10.93	0.6896
25–29	105	12	11.43	0.5362	237	35	14.77	**0.0128**
30–34	137	17	12.41	0.2576	164	17	10.37	0.9477
35–39	154	13	8.44	0.5615	196	21	10.71	0.8077
40–44	98	9	9.18	0.8491	135	17	12.89	0.3430
45–49	70	6	8.57	0.7359	72	13	18.06	**0.0249**
50–54	74	8	10.81	0.7447	56	3	5.36	0.2222
55–59	53	3	5.66	0.3058	40	3	7.50	0.5658
60–64	41	2	4.88	0.4208	41	3	7.32	0.7931
65–69	30	2	6.67	0.7609	32	2	6.25	0.7657
70–74	17	1	5.88	1.0000	14	1	7.14	1.0000
75–79	10	0	0	0.6109	10	0	0	0.6116
80–84	3	1	33.33	0.2646	0	0	0	-
85–89	1	0	0	1.0000	2	0	0	1.0000

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
