# Peer review of "Risk of Postoperative Bleeding in Tonsillectomy for Peritonsillar Abscess, as Opposed to in Recurrent and Chronic Tonsillitis—A Retrospective Study"

_ijerph, 2021, doi:10.3390/ijerph18041946_

Round 1

Reviewer 1 Report

The aim of the study can be describe exactly

The conclusions are very poor. The results can improved the impact to clinical practice. 

The authors can described the aim of the study. I mean the description of impact of parametres where were analsyed. The question is what parametr we can improve in the patients care schedule. The conclusions must be real for clinical practice , that conclusion depend the analyse and results of study and can describe the impact to recommandation for any improving in treatment and/or postop care protocols. I belive that the study is very good in designe and analytics part and would be improve the daily clinical practice for ENT departments worldwide. This information are important and the conclusions and recommandation are important for many ENT departments.

Author Response

Review 1

Open Review

         English language and style

( ) Extensive editing of English language and style required
( ) Moderate English changes required
(x) English language and style are fine/minor spell check required
( ) I don't feel qualified to judge about the English language and style

Yes

Can be improved

Must be improved

Not applicable

Does the introduction provide sufficient background and include all relevant references?

(x)

( )

( )

( )

Is the research design appropriate?

( )

(x)

( )

( )

Are the methods adequately described?

(x)

( )

( )

( )

Are the results clearly presented?

(x)

( )

( )

( )

Are the conclusions supported by the results?

( )

(x)

( )

( )

Comments and Suggestions for Authors

Changes made based on the editor’s suggestions are highlighted green in the manuscript and changes made based on the reviewers’ comments were highlighted yellow.

Point 1: The aim of the study can be described exactly

Response 1:

  • Accepted, changed and added in manuscript – lines 63-66, 244-2466:

The aim of the study was to determine whether the indication for acute tonsillectomy in PTA (different pathophysiological conditions - oedema, acute inflammation in tissue, more difficult orientation in the surgery field) is associated with a higher risk of postoperative bleeding than elective tonsillectomy indicated for CHT, or RT.

  • Explanation was merged with the response to instruction „The authors can described the aim of the study“, two lines below.
  • The surgery in inflammatory field is riskier than in uninflammed tissue. The major aim of our study was to explore if the more complicated surgery of tonsillectomy for PTA has an impact on the occurence of PTH if we compare it to the group indicated for CH and RT. A secondary or minor aim was to assess if there is an impact of age, gender, season and side of surgery and find the riskiest days for PTH because we assumed that the riskiest days fall into the home-care period of the posttonsillectomy care. PTH occurred in 60,7% of our sample of patients in home-care period.

Point 2: The conclusions are very poor. The results can improve the impact to clinical practice. 

  • Response 2:Accepted, changed and added in manuscript – line 299-305:

Tonsillectomy is undoubtedly a routine operation in otorhinolaryngology; the occurrence of postoperative bleeding is not, however, a rare complication. Techniques to minimize the risk of bleeding have not yet been found. Despite the completely different conditions in the field of surgery in tonsillectomy for PTA, our work demonstrated an almost identical risk of postoperative bleeding in the indication for tonsillectomy as a treatment of PTA as in the indication for CHT+RT. Among the biggest risks resulting from tonsillectomy is the fact that the riskiest days of PTH fall into the home-care period of post-tonsillectomy treatment. In our sample, PTH occurs in more than 60% of patients in the home environment

The sentence: “A certain limit of our study may be its monocentricity. The study’s major advantage is its large group of operated patients (2842 tonsillectomy for a period of 4 years).” was moved to discussion – line 294-295.

Point 3: The authors can describe the aim of the study.

Response 3:

  • Accepted, changed and added above.

Point 4: I mean the description of impact of parametres where were analsyed. The question is what parametr we can improve in the patients’ care schedule. The conclusions must be real for clinical practice , that conclusion depend the analyse and results of study and can describe the impact to recommendation for any improving in treatment and/or postop care protocols.

Response 7:

  • It is really difficult to find a parametr and improve care schedule because all the ENT doctors in tonsillectomy have to balance economical considerations (shorter hospitalization is cheaper) with medical factors (riskiest days are between 5. and 8. postoperative day). We believe that for common ENT practise it is useful to know that (surprisingly) despite the completely different conditions in the fields of surgery (oedema, acute inflammation in peritonsillar abscess), there was no statistically significant difference between incidence of PTH in the studied groups. Informing the patients that PTH occurs in more than 60% of patients during  home-care period of treatment is useful for our patients.

Point 5: I belive that the study is very good in designe and analytics part and would be improve the daily clinical practice for ENT departments worldwide. This information are important and the conclusions and recommandation are important for many ENT departments.

Response 5:

  • We also believe that our repaired results and conclusions bring a new point of view on the incidence of PTH in PTA compared to CHT+RT to ENT clinical practise. Equally, that patients can be informed that PTH occurs in more than 60% of patients during the home-care period is useful for encouraging patient cooperation and communication.

Reviewer 2 Report

Authors tried to evaluate risk factors of post-tonsillectomy bleeding, and compared it according to the indication of tonsillectomy.

This subject is worthy, and need to be investigated.

However there are several issues in this manuscript.

Major

-This is a retrospective study, and how did authors distinguish the indication of tonsillectomy? Reviewing medical record of 1994, might not be helpful in some cases. Furthermore, snoring/obstructive sleep apnea is one of most common indications for tonsillectomy. How many cases were enrolled or excluded? (As this study includes pediatrics, snoring could be more common indication of tonsillectomy than peritonsillar abscess/recurrent tonsillitis, especially in pediatrics).

Authors should clearly demonstrate the methodology about chart review, and inclusion/exclusion criteria needs to be specifically described. As this is a retrospective study including relative large population for a long duration, adding a graphic demonstration of subjects’ enroll/exclusion process would be helpful (as a figure 1 or supplementary figure).

-In line 75, authors’ description of table 1 is very hard to understand. ‘894 patients in total and 1742 performed tonsillectomies’ means ‘some patients underwent bilateral tonsillectomy, and other underwent unilateral tonsillectomy, therefore total number of resected tonsils is 1742’ ? I understood that all group 1 subjects underwent bilateral tonsillectomy, and group 2 is consist of patients who underwent unilateral and bilateral tonsillectomy. More details need to be described.

-Simillarly, in abstract section, ‘2842 tonsillectomies’ means ‘2842 resected tonsils’?

‘Bleeding occurred in 285 cases’ means ‘285 patients with bleeding’? (Then, what is the number of tonsils with bleeding?)

The description of ‘tonsillectomies’, ‘cases’ are confusing. It needs to be corrected in whole manuscript.

-There are many factors that affect postoperative bleeding. They could be grouped as pre-operative, intraoperative and postoperative thing. Patients’ characteristics such as age, gender, concurrent medical conditions (for examples, presence of hypertension, diabetes) are pre-operative characteristics. Operation time, intra-operative bleeding amount, intra-operative medications could be intraoperative things. Postoperative analgesics, patients’ diet could be post-operative factors. These factors should be considered together to suggest true risk factors of post-tonsillectomy bleeding. If authors are able to further evaluate risk factors according to pre-, intra-, post-operative factors, it would be more helpful.

-Who performed operation? As this study involved relatively longer duration, information about surgeon’s factor should be considered.

Minor

-In line 53, CHR should be corrected as CHT.

Author Response

Review 2

Open Review

         English language and style

( ) Extensive editing of English language and style required
( ) Moderate English changes required
( ) English language and style are fine/minor spell check required
(x) I don't feel qualified to judge about the English language and style

Yes

Can be improved

Must be improved

Not applicable

Does the introduction provide sufficient background and include all relevant references?

(x)

( )

( )

( )

Is the research design appropriate?

( )

( )

( )

(x)

Are the methods adequately described?

( )

( )

( )

(x)

Are the results clearly presented?

( )

( )

(x)

( )

Are the conclusions supported by the results?

( )

( )

(x)

( )

Comments and Suggestions for Authors

Changes made based on the editor’s suggestions are highlighted green in the manuscript and changes made based on the reviewers’ comments were highlighted yellow.

Authors tried to evaluate risk factors of post-tonsillectomy bleeding, and compared them according to the indication of tonsillectomy.

This subject is worthy, and need to be investigated.

However there are several issues in this manuscript.

 Major

Point 1: This is a retrospective study, and how did authors distinguish the indication of tonsillectomy? Reviewing medical record of 1994, might not be helpful in some cases.

Response 1:

- In our department, patient history and the indication for tonsillectomy is the standard part of patients’ report form and in the years included in the study (2014-2019) our documentation was in electronic form. Every patients icluded in the study was checked for the indication of TE.

Point 2: Furthermore, snoring/obstructive sleep apnea is one of most common indications for tonsillectomy. How many cases were enrolled or excluded? (As this study includes pediatrics, snoring could be more common indication of tonsillectomy than peritonsillar abscess/recurrent tonsillitis, especially in pediatrics).

Response 2:

- Only patients with indication of PTA, CHT or RT were enrolled in this study. Patients indicated for TE for OSA were excluded (see exclusion criteria line 104-111). It was a sample of 51 patients (ie.104 tonsillectomies). The reason for this decision was based on fact that in our country TE for snoring/OSA in children is in most cases  part of „adenotonsillectomy“ (recommendations of the board of Otorhinolaryngology and Head and Neck Surgery) and in adults very often TE is part of UPPP or it is combined with RFITT of radicis linguae. TE only patients were very rare and so it was simpler to exclude all of them.

Point 3: Authors should clearly demonstrate the methodology about chart review, and inclusion/exclusion criteria needs to be specifically described. As this is a retrospective study including relative large population for a long duration, adding a graphic demonstration of subjects’ enroll/exclusion process would be helpful (as a figure 1 or supplementary figure).

Response 3:

- The sample of tonsillectomies performed in our department in 2014-2019 comprises 2340 patients. It containes TE (except PTA, CHT, RT) indicated for OSA, diagnostic TE, oncological indications, processus styloideus elongatus, trauma etc. In these other indications the field of surgery can be affected. The study was focused on the main indications for TE. In our study PTA, CHT and RT comprise more than 90% of the indications.

The main aim of the study was to determine whether the indication for acute tonsillectomy in PTA (many different pathophysiological conditions - oedema, acute inflammation in tissue, more difficult orientation in the surgery field,) is associated with a higher risk of postoperative bleeding than elective tonsillectomy indicated for CHT, or RT.

- For this purpose, the description and analysis of the all sample of TE in our department seemed to be expendable and without clinical benefit.

- The choice of PTA versus CHT+RT allowed a comparison of a clearly defined groups: with pathophysiological differences and without.

Point 4: In line 75, authors’ description of table 1 is very hard to understand. ‘894 patients in total and 1742 performed tonsillectomies’ means ‘some patients underwent bilateral tonsillectomy, and other underwent unilateral tonsillectomy, therefore total number of resected tonsils is 1742’ ? I understood that all group 1 subjects underwent bilateral tonsillectomy, and group 2 is consist of patients who underwent unilateral and bilateral tonsillectomy. More details need to be described.

Response 4:

- Accepted, changed and added in manuscript – line 80-83:

Details of the demographic distribution of the file are given in Table 1. There were 1,100 patients in group I., in group II. there were 894 patients in total (650 patients: 276 men (42,46%), 374 women (57,54%) were indicated for CHT; 244 patients: 96 men (39,34%), 148 women (60,66%) were indicated for RT).

-and line 87-102:

For the purposes of the study, the data of a total of 2842 unilateral tonsillectomies (resected tonsils) were included in the study group, of which 1146 tonsillectomies were performed primarily as a unilateral procedure and 1742 tonsillectomies were performed as part of bilateral procedures.

There were 1,100 patients in group I. and all of these patients underwent unilateral tonsillectomy on the abscess-side; Group I. contains 1100 unilateral tonsillectomies (1100 resected tonsils; 38,71% of the whole study).

In group II. there were 894 patients in total and 1742 resected tonsils (61,29% of the whole study).

650 patients: 276 men (42,46%), 374 women (57,54%) were indicated for CHT in group II. The sample included 1262 resected tonsils (44,40% of the whole study).

244 patients: 96 men (39,34%), 148 women (60,66%) were indicated for RT in group II., and it comprised 480 resected tonsils (16,89% of the whole cohort).

Type of surgery distribution is presented in Table 2.

Point 5: Simillarly, in abstract section, ‘2842 tonsillectomies’ means ‘2842 resected tonsils’?

Response 5:

- Accepted, changed and added in manuscript – line  27-28:

…makes this issue topical. 2842 unilateral tonsillectomies from the years 2014-2019 were included…

Point 6: Bleeding occurred in 285 cases’ means ‘285 patients with bleeding’? (Then, what is the number of tonsils with bleeding?)

Response 6:

- The number of cases and the number of tonsils with bleeding is the same because in our study there was no patient with bleeding on both sides.

Point 7: The description of ‘tonsillectomies’, ‘cases’ are confusing. It needs to be corrected in whole manuscript.

Response 7:

- Accepted, changed in all manuscript

- „case“ –removed line: 28,

-left, this makes sense, used in general meaning line 59, 110, 118, 119, 183, 185, 186, 190, 203, 236,

- left, this is a title in References line 324-374

 -„tonsillectom..“ -left, this makes sense, used in general meaning line 2, 20, 23, 25, 26, 27, 28, 40, 41, 50, 52, 56, 63, 66, 74, 88, 89, 90, 93,103, 104, 106, 112, 184, 190, 198, 202, 206, 214, 223,233, 237, 244, 248,  250, 251, 252, 269, 298, 301, 302, 303, 304

-left and explained line 89

-removed and changed line 94, 97

- left, this is a title in References line 315-422

Point 8: There are many factors that affect postoperative bleeding. They could be grouped as pre-operative, intraoperative and postoperative thing. Patients’ characteristics such as age, gender, concurrent medical conditions (for examples, presence of hypertension, diabetes) are pre-operative characteristics. Operation time, intra-operative bleeding amount, intra-operative medications could be intraoperative things. Postoperative analgesics, patients’ diet could be post-operative factors. These factors should be considered together to suggest true risk factors of post-tonsillectomy bleeding. If authors are able to further evaluate risk factors according to pre-, intra-, post-operative factors, it would be more helpful.

Response 8:

-Comparable published studies divide the factors of PTH in many manners. We used 2 samples of unselected patients from common work in a university hospital. All of the patients included were admitted to surgery only with well compensated comorbidities. In the other case in group I. (with acute TE), they were treated by peritonsilar incision (very rare treatment today), if it was neccesary. In group II. there were only elcktive surgeries. Published studies had conflicting conclusions on the topic for hypertension or diabetes and PTH. In our opinion, the most beneficial studies concluded that a higher risk of PTH is conected with systolic blood pressure over 140 mmHg/torr. We agree with this conclusion:  these patients are not sufficiently compensated for the purposes of tonsillectomy. The similar situations in principle have been described in studies on the topic of diabetes.

-Operation time was not monitored in this study because PTA versus CHT+RT are procedures with different conditions in the surgery field.

- The intra-operative bleeding amount was not monitored because fundamental decrease of hemoglobin or other markers requiring intesive care were not found in any PTH patient. We think that it is a nice result and of importance for our work.

- Postoperative pain management is based on paracetamol and metamizol. Available studies proved these drugs are not conected with PTH.

- That is true, diet folowing TE can be factor for PTH. In our department all of our patient have a standardized diet. All of them are educated by the medical staff on this topic in the home-care period, but we can not influence their diets at home. We did not monitor it because data are not correct.

- We believe that the current design of our study would be beneficial for our target. We have tried to accept all of other suggestions.

Point 9: Who performed operation? As this study involved relatively longer duration, information about surgeon’s factor should be considered.

Response 9:

- We know and checked studies focused on surgeon´s factor. In our hospital and in this study performing doctors are working independently when they are board certified. The others are working under supervision of board-certified doctor.

Minor

Point 10: In line 53, CHR should be corrected as CHT.

Response 10:

- Accepted, changed – line 55

Reviewer 3 Report

For the clinician idea behind the study is interesting because it is often assumed that inflammation is a risk factor for postoperative bleeding after "tonsillectomy à chaud" - tonsillectomy with abscess drainage.

According to the abstract the authors claim to have done a retrospective study with 2842 tonsillectomies in 1994 pts: No dropouts! No mentioning: lost to follow up! No data about how many excluded pts! How the authors could be sure that everyone with a postoperative bleeding (even minor) joined their hospital?

In the methods section  basically only the fact of postoperative bleeding is reported, not the degree. As described seemingly everyone with the slightest bleeding was hospitalized. How long on average?

Which intervention in how many cases?

Degree of bleeding? Hb?

The major issue is this than: there is no information about the severeness of bleeding. This gives doubt about the reliability of the results: it could be that the incidence of bleedings is as the authors report - without a significant difference in quantity - but a significant difference in quality of bleeding - which could be hidden behind the data of the incidence. This should be ruled out - and a resubmission could be possible.

Author Response

Review 3

Open Review

         English language and style

(x) Extensive editing of English language and style required
( ) Moderate English changes required
( ) English language and style are fine/minor spell check required
( ) I don't feel qualified to judge about the English language and style

Yes

Can be improved

Must be improved

Not applicable

Does the introduction provide sufficient background and include all relevant references?

( )

( )

(x)

( )

Is the research design appropriate?

( )

( )

(x)

( )

Are the methods adequately described?

( )

( )

(x)

( )

Are the results clearly presented?

( )

(x)

( )

( )

Are the conclusions supported by the results?

( )

( )

(x)

( )

Comments and Suggestions for Authors

Changes made based on the editor’s suggestions are highlighted green in the manuscript and changes made based on the reviewers’ comments were highlighted yellow.

For the clinician idea behind the study is interesting because it is often assumed that inflammation is a risk factor for postoperative bleeding after "tonsillectomy à chaud" - tonsillectomy with abscess drainage.

Point 1: According to the abstract the authors claim to have done a retrospective study with 2842 tonsillectomies in 1994 pts: No dropouts! No mentioning: lost to follow up! No data about how many excluded pts!

Response 1:

- Patients with uncomplicated courses of healing were monitored by an outpatient ENT and their GP once a week during the home-care period, with the GP deciding when they could go back to work. The total duration of sick leave and follow up lasted two to three weeks. The patients in the PTH group were individually monitored in the home-care period in the outpatient consulting room of the ENT department where the surgery was done. The maximum duration of follow up was 5 weeks.

- Patients who cooperated badly (for example signed a waiver in conflict to doctor´s redommendation, smoking after surgery, sport practising, traveling to a congress) were excluded (n=65).

Point 2: How the authors could be sure that everyone with a postoperative bleeding (even minor) joined their hospital?

Response 2:

- The region of our university hospital is without an adjacent ENT department; there is only an outpatient ENT care available. The standard management of PTH is admitting the patient to hospital with every PTH (including anamnestic). So we suppose, that we have included all the cases of PTH in this study.

Point 3: In the methods section  basically only the fact of postoperative bleeding is reported, not the degree. As described seemingly everyone with the slightest bleeding was hospitalized. How long on average?

Response 3:

- To divide bleeding to degree in emergent situation like PTH is always in poor objectivity. That´s why we divide it by the necessary treatment to “working practical degree of „anamnestic and minor insignificant bleeding“ – observation and hemostyptics, to degree of „ significant bleeding“ - tonsillar fossa compression using a gauze tampon and bipolar coagulation under local anesthesia were used to control it and degree of „unsuccessful treatment in local anesthesia or of massive bleeding”- electrocoagulation under general anesthesia (other methods - suture of palatal arches, hypopharyngeal tamponade - were not used). Line 113-119 (the part line 113-119 is green because there are accepted editors suggestions too):

All patients with postoperative bleeding were hospitalized, even if the data was only anamnestic. Anamnestic and minor insignificant bleeding were addressed by observation and application of hemostyptics. Tonsillar fossa compression using a gauze tampon and bipolar coagulation under local anesthesia were used to control significant bleeding. Electrocoagulation under general anesthesia was indicated in cases of unsuccessful treatment of bleeding with local anesthesia or in cases of massive bleeding. Other methods (suture of palatal arches, hypopharyngeal tamponade) were not used.

Point 4: Which intervention in how many cases?

Response 4:

- It is counted and forgot it to include to manuscript. Accepted and added in manuscript – line 182-185:

- From the total number of the first PTH (n=285), a second PTH occurs in 32 cases (11,23%). In the PTA group, 11 tonsillectomies i.e.,11,28%; in the CH+RT group, 21 tonsillectomies, i.e., 11,80%. A third PTH occurs in 3 cases (1,05%).: 1 case (0,93%) in group I.; 2 cases in group II.(1,12%). Without significant differences in both groups (p=0,5774).

Point 5: Degree of bleeding? Hb?

Response 5:

- Degree of bleeding was discused above. Post-operative level of hemoglobin was monitored in cases with PTH. The main reason for us not discussing hemoglobin was that there were no cases requiring intensive care (cases of dramatic, life-threatening bleeding requiring blood replacement).

Point 6: The major issue is this than: there is no information about the severeness of bleeding. This gives doubt about the reliability of the results: it could be that the incidence of bleedings is as the authors report - without a significant difference in quantity - but a significant difference in quality of bleeding - which could be hidden behind the data of the incidence. This should be ruled out - and a resubmission could be possible.

Response 6:

- As we declared above, we monitored and divided bleeding to degree in view of necessary management of bleeding – by the highest used care. I.degree was treated by observation and hemostyptics: anamnestic minor insignificant bleeding. II.degree was treated in local anesthesia by compression using a gauze tampon or bipolar coagulation. III.degree was treated in general anesthesia (electrocoagulation for massive bleeding or unsuccessful treatment in local anesthesia).

- As for the severity of bleeding, we monitored parameters (hemoglobin etc.) but no case required blood transfusion, or any such measures. That is the reason why we did not discuss it in detalis.

Round 2

Reviewer 2 Report

Authors tried to improve the manuscript following all reviewers' suggestions.

The limitation of the study that 'the post-operative diet was not evaluated' needs be added as a limitation of the study in manuscript.

Reviewer 3 Report

After spending some time carefully reading the answers of the authors to the reviewers questions, I opened the manuscript to find out that only a part of the answers were integrated and complemented. To make it clear: the recommendations and questions are not to help the reviewer to understand but to improve the manuscript in terms of scientific clarity and reproducibility.

Eg. when pts were excluded it should be mentioned with their quantity and  reason for exclusion in the methods section: not done - this is only 1 example.

Anothe example: Response 6:

- As we declared above, we monitored and divided bleeding to degree in view of necessary management of bleeding – by the highest used care. I.degree was treated by observation and hemostyptics: anamnestic minor insignificant bleeding. II.degree was treated in local anesthesia by compression using a gauze tampon or bipolar coagulation. III.degree was treated in general anesthesia (electrocoagulation for massive bleeding or unsuccessful treatment in local anesthesia).

  • As for the severity of bleeding, we monitored parameters (hemoglobin etc.) but no case required blood transfusion, or any such measures. That is the reason why we did not discuss it in detalis.

Where is it to be found in the manuscript, eg. a table? an annotation?
